# Hydrophobin CmHYD1 Is Involved in Conidiation, Infection and Primordium Formation, and Regulated by GATA Transcription Factor CmAreA in Edible Fungus, *Cordyceps militaris*

**DOI:** 10.3390/jof7080674

**Published:** 2021-08-20

**Authors:** Xiao Li, Fen Wang, Mengqian Liu, Caihong Dong

**Affiliations:** 1State Key Laboratory of Mycology, Institute of Microbiology, Chinese Academy of Sciences, Beijing 100101, China; lixmushroom@gmail.com (X.L.); wangfen@im.ac.cn (F.W.); liumengqian1011@gmail.com (M.L.); 2University of Chinese Academy of Sciences, Beijing 100049, China; 3Guizhou Key Laboratory of Edible Fungi Breeding, Guizhou Academy of Agricultural Sciences, Guiyang 550000, China

**Keywords:** *Cordyceps militaris*, *Cmhyd1*, conidiation, hydrophobicity, infection, primordium differentiation, GATA transcription factor AreA, positive feedback

## Abstract

Hydrophobins are a family of small proteins exclusively secreted by fungi, and play a variety of roles in the life cycle. *Cmhyd1*, one of the hydrophobin class II members in *Cordyceps militaris*, has been shown to have a high transcript level during fruiting body development. Here, deletion of *Cmhyd1* results in reduction in aerial mycelia, conidiation, hydrophobicity and infection ability, and complete inhibition of pigmentation and primordium differentiation. *Cmhyd1* plays roles in conidiation and cuticle-bypassing infection by regulating the transcripts of frequency clock protein, *Cmfrq*, and velvet protein, *Cmvosa*, as well as primordium formation via the mitogen-activated protein kinase signaling pathway. *Cmhyd1* also participates in stress response, including tolerance of mycelia to osmotic and oxidative stresses, and conidia to high or low temperatures. CmAreA, a transcription factor of nitrogen regulatory, is recruited to the promoter of *Cmhyd1* and activates the transcription of *Cmhyd1* with coactivator CmOTam using electrophoretic mobility shift assays and transient luciferase expression in tobacco. Furthermore, CmHYD1 is proved to regulate the transcription of *Cmarea* at different developmental stages via a positive feedback loop. These results reveal the diverse roles and regulation of *Cmhyd1* in *C. militaris,* and provide insights into the developmental regulatory mechanism of mushrooms.

## 1. Introduction

Hydrophobins are small cysteine-rich amphiphilic proteins produced exclusively by fungi [1]. They have traditionally been grouped into class I and II based on their hydropathy patterns and solubility characteristics [2,3]. They are secreted proteins which react to interfaces between fungal cell walls and the air, or between fungal cell walls and solid surfaces, by self-assembling into insoluble polymerized amphipathic monolayers [4].

Hydrophobins have been characterized in some fungi, especially pathogenic fungi of plants and insects. In the rice blast fungus *Magnaporthe grisea* (Magnaporthales), hydrophobins were necessary for fungal development, infection ability and plant colonization [5,6,7]. In the entomopathogenic fungus *Beauveria*
*bassiana* (Hypocreales), the inactivation of *hyd1* resulted in reduced spore hydrophobicity and fungal virulence, but a Δ*hyd2* mutant displayed decreased surface hydrophobicity of conidia without effects on virulence [8]. In another entomopathogenic fungus, *Metarhizium brunneum* (Hypocreales), hydrophobins played roles in conidiation, hydrophobicity, pigmentation and virulence [9]. *HYD3*, a hydrophobin located on the conidial surface of *M. acridum* (Hypocreales), specifically activated the humoral and cellular immunity of its own host insects [10]. Hydrophobins were required for conidial hydrophobicity and plant root colonization in the fungal biocontrol agent *Clonostachys rosea* (Hypocreales) [11]. In mycoparasite fungi *Trichoderma longibrachiatum* (Hypocreales), hydrophobins affected hydrophobicity of conidia, disease resistance, pathogenicity and plant growth promotion activity [12]. Contrary to these reports, individual deletion of the five hydrophobin genes (*HYD1*-*5*) in *Fusarium verticillioides* (Hypocreales) indicated that none were required for virulence in a corn seedling infection assay [13]. Similar results were reported for *Botrytis cinerea* (Helotiales) in which none of the hydrophobin mutants showed obvious phenotypic defects [14]. It seemed that the function of hydrophobic proteins was species specific.

Though the first characterized hydrophobin was *SC3* from mushroom-forming fungus *Schizophyllum commune* (Agaricales) [15], hydrophobins in mushrooms are far from well studied. It was reported that a hydrophobin gene, *Hyd9*, played an important role in the formation of aerial mycelia and primordia in *Flammulina filiformis* (Agaricales) by RNAi and overexpression [16]. Hydrophobin *FBH1* in *Pleurotus ostreatus* (Agaricales) affected the mycelial growth rate and primordium formation by RNAi transformants [17]. There have only been some studies on the gene transcript level of hydrophobin genes. Banerjee et al. [18] reported that hydrophobins *SC3* and *SC4* are expressed in mounds, fruiting bodies and vegetative hyphae of *S. commune*. The transcript level of *fv-hyd1* in *F. filiformis* markedly increased at the primordium stage, as shown by Northern blot [19]. In *Tricholoma vaccinum* (Agaricales), expression levels of nine hydrophobin genes throughout the life cycle revealed their essential role in aerial mycelium, fruiting body and ectomycorrhiza establishment [20].

*Cordyceps militaris*, one of the entomopathogenic fungi, is different from species of the genera *Beauveria* and *Metarhizium*. It can produce fruiting bodies on both host silkworm pupae and wheat medium [21,22] and the fruiting bodies have been used as food and nutrition tonics worldwide, especially in eastern Asia. Furthermore, this fruiting body-forming fungus is genetically tractable, which enables the identification and characterization of a number of genes controlling development in *C. militaris*.

Our previous study reported that there were four hydrophobin-encoding genes in *C. militaris* according to the results of domain structure, hydropathy pattern and phylogeny analysis. CmHYD1 and CmHYD2 were members of class II, while CmHYD3 and CmHYD4 were members of class I. Phylogenetic analysis based on hydrophobin proteins of *Cordyceps* s.l. revealed more variability among the class II members, and there was only 50% similarity between CmHYD1 and the phylogenetically close species *B. bassiana* (PMB64752.1). Each hydrophobin gene participated in different developmental processes during the life cycle, revealed by transcript profiling [23]. *Cmhyd1*, one of the hydrophobin class II members, showed a high transcript level during the fruiting body development compared with the mycelial stage when *C. militaris* was cultured on both insects and wheat media [23]. However, the function of *Cmhyd1* during the whole life cycle of *C. militaris* remains to be verified.

Since the important and diverse roles of hydrophobins for some fungi have been reported, how the hydrophobin genes are regulated is our concern. However, there is little information in *Ma.*
*grisea*, *Me.*
*anisopliae* and *Me. robertsii*. NPR1 and NPR2 (nitrogen pathogenicity regulation) are wide-domain regulators of nitrogen metabolism required for pathogenicity and induction of *MPG1* (hydrophobin gene) expression during nutrient starvation in *Ma.*
*grisea*, based on Northern blot experiments [24]. Further studies showed that the PMK1 MAPkinase pathway, the cAMP-response pathway and the wide domain regulators of nitrogen source utilization regulated the full expression of *MPG1* during spore formation and plant infection [25]. A regulator of a G protein signaling (RGS) gene, *cag8*, was involved in hydrophobin synthesis in the insect-pathogenic fungus *Me.*
*anisopliae* [26]. In *Me. robertsii*, a zinc finger transcript factor MrCre1, termed a carbon catabolite repressor, was confirmed to regulate *Mrhyd4,* a class I hydrophobin, to modulate infection structure (appressorium) formation and virulence [27,28]. It was assumed that a nutrition-linked regulatory mechanism may be involved in the regulation of hydrophobin genes.

In the present study, *Cmhyd1* was characterized in detail in *C. militaris*. *Cmhyd1* was found to be involved in conidiation, pigmentation, infection, primordium formation and stress responses. To investigate how the hydrophobin gene *Cmhyd1* is regulated, a core transcription factor of nitrogen regulation, CmAreA, was confirmed to be recruited to the promoter and activate the transcript of *Cmhyd1* in vivo and in vitro. Furthermore, it was found that CmHYD1 can regulate the transcript level of *Cmarea*. Our results revealed the diverse roles of *Cmhyd1*- and *Cmarea*-associated regulatory pathways in hydrophobin genes for the growth and development of the edible fungus *C. militaris*.

## 2. Materials and Methods

### 2.1. Strains and Culture Conditions

The *C. militaris* strain CGMCC 3.16323 was maintained on potato dextrose agar (PDA) at 20 °C. The *Escherichia coli* strain DH5α (Tiangen Biotech Co., Ltd., Beijing, China) and *Agrobacterium tumefaciens* strains AGL-1 (Tiangen Biotech Co., Ltd., Beijing, China) and GV3101 (Beijing Zoman Biotech Co., Ltd., Beijing, China) used for DNA manipulation and fungal transformation were stored at −80 °C. *E. coli* strains were cultured with Luria–Bertani medium supplemented with spectinomycin (100 μg/mL) or kanamycin (50 μg/mL) for selection markers. *A. tumefaciens* strains were cultured with yeast extract beef (YEB) medium supplemented with carbenicillin (50 μg/mL) and kanamycin (50 μg/mL) for strain AGL-1 and with rifampicin (100 μg/mL), spectinomycin (100 μg/mL) or kanamycin (50 μg/mL) for strain GV3101.

For phenotype observation, the strains were grown under constant dark at 20 °C for 21 d and then transferred to 12 h dark:12 h light at 20 °C for 2 or 4 d. Liquid cultures were grown in media with 200 g/L potato, 20 g/L glucose, 3 g/L peptone, 1 g/L KH_2_PO_4_, 0.5 g/L MgSO_4_. The diameters of colonies on PDA medium were measured to evaluate the growth rates by the cross-line method. The conidial production was determined after being cultured under 12 h dark:12 h light at 20 °C for 4 d. The conidia were harvested by adding 10 mL ddH_2_O to the PDA plate, mixing fully and filtering through sterilized gauze to remove the mycelia. Finally, the conidia were counted in a hemocytometer. The obtained data represented three biological replicates, with two technical replicates each.

### 2.2. Gene Deletion, Complementation and Overexpression

All the molecular operations were done with a CloneExpress^®^ Ultra One Step Cloning Kit (C115, Vazyme Biotech Co., Ltd., Beijing, China). The ORF cassettes of *Cmhyd1* and *Cmarea* were obtained from the genome of *C. militaris* strain CM01 (accession: SRA047932) [29]. The 5′- and 3′-flanking regions of *Cmhyd1* (927 and 889 bp, respectively) and *Cmarea* (1180 and 1019 bp, respectively) were amplified from the genomic DNA of the wild type (WT) strain. The 5′-flanking regions, hygromycin resistance gene (*Hyg*) cassette driven by a *trpC* promotor and 3′-flanking regions were cloned into vector pAg1-H3 (provided by Prof. Xingzhong Liu from Nankai University) to generate the deletion vectors pAg1-hyg-*Cmhyd1* and pAg1-hyg-*Cmarea*. Mutant strains obtained by *A. tumefaciens*-mediated transformation were verified by polymerase chain reaction (PCR) and reverse transcription PCR (RT-PCR).

Gene complementation was carried out with *Cmhyd1* (1165 bp) or *Cmarea* (4142 bp) full-length coding sequences, including the original promoter and terminator regions which were ligated into pAg1-bar (this lab) to generate the complementation vectors pAg1-bar-*Cmhyd1* and pAg1-bar-*Cmarea*. Δ*Cmhyd1* and Δ*Cmarea* strains were used for gene complementation. The complementation transformants were screened on PPDA medium with 2 mg/mL glufosinate (bar) and then verified by PCR.

For gene overexpression, the strong promoter region of *Cmgpd* from *C. militaris* and the ORF sequence of *Cmhyd1* (445 bp) as well as terminator *TtrpC* from *A. nidulans* were ligated into vector pAg1-H3 to generate the overexpression vector pAg1-hyg-P*_Cmgp_*-*Cmhyd1*. The screened overexpression mutants were labeled as *Cmhyd1oe-2* and *Cmhyd1oe-4*. All the primer sequences used in this study are listed in Appendix A.

### 2.3. Determination of the Hydrophobicity of the Mycelia and Conidia

Twenty microliters of 0.1% Tween 80 was added on the surface of the mycelia. The diffusion capacity of 0.1% Tween 80 drop was observed within 30 min. The conidial suspensions containing 0.1% Tween 80 solution were observed by light microscope to check their hydrophobicity. An alternative method used for comparing the hydrophobicity of the conidia is the microbial adhesion to hydrocarbons (MATH) assay [30]. Conidia were suspended in a PBS/hexadecane mixture and shaken vigorously. Most of the conidia attached to the oil–buffer interface. The conidia can be released from this interface by adding the detergent Triton X-100, and titrating with the detergent reveals quantifiable differences in conidium hydrophobicity between strains.

### 2.4. Observation of the Hyphae and Conidia by Light Microscope and Scanning Electron Microscope 

The WT and mutant strains were grown under constant dark at 20 °C for 21 d and then transferred to 12 h dark:12 h light at 20 °C for 2 d. The hyphae and conidia were observed by light microscope and scanning electron microscope (SEM) (Hitachi SU8010, Tokyo, Japan). For the SEM, mycelia of about 0.5 cm^2^ with an agar disc were fixated with 2.5% glutaraldehyde (McLean Biochemical Technology, Shanghai, China) at 4 °C for 24 h. The glutaraldehyde was removed and the samples were eluted with deionized water three times. Subsequently, the deionized water was removed and samples were eluted with 70%, 85%, 95%, and 100% ethanol, respectively. Finally, the sample were glued on stubs, covered with a thin conductive layer and observed by SEM after critical-point drying and gold sputtering [31].

### 2.5. Stress Adaptation Assays

The effects of osmotic and oxidative stresses on the growth were determined on PDA media supplemented with cell wall-damaging agent SDS and Congo red (CR), and an oxidative stress agent (H_2_O_2_). Agar plugs with colonized WT and all the mutants were inoculated on PDA plates containing 10% CR, 0.1% SDS and 0.075 mM H_2_O_2_, respectively. Colony diameters were then measured after growing under constant dark at 20 °C for 21 d.

For the effect of temperature stress on conidium germination, a 100 μL conidium suspension with a concentration of 10^5^/mL was treated at −80, −20, 0, 35, 40 and 45 °C for 0, 0.5, 1.0, 1.5, 2.0, 2.5 and 3.0 h, respectively. Then, 1 mL PDB media was added and samples were cultured at 20 °C and 200 rpm for 24 h. The germination of conidia was observed under a light microscope and counted in a hemocytometer.

### 2.6. Heterologous Expression of CmHYD1 in Pichia pastoris and Feeding

CmHYD1 was expressed in *P. pastoris* by cloning the cDNA sequence of *Cmhyd1* into vector pPICZαA (provided by Prof. Gang Liu from the Institute of Microbiology, Chinese Academy of Sciences) with an additional C-terminal 6×His-tag as described by Stübner et al. [32]. Purified CmHYD1 at a concentration of 200 mg/mL was added to the wheat media after scratching for fruiting body cultivation or mixed with blastospores to infect silkworm pupae.

### 2.7. Assays for Fungal Virulence

One hundred microliters of a 10^7^ conidia/mL suspension with or without purified CmHYD1 was injected into each pupa in three groups (6 per group) for cuticle-bypassing infection. All treated groups were maintained at 20 °C and monitored daily for infection until all the silkworm pupae were completely mummified.

### 2.8. Fruiting Body Production in Cordyceps militaris 

All strains for fruiting were cultivated in wheat medium as previously described [22]. Fruiting bodies were harvested and the fresh weights were recorded. Biological efficiency (BE) for each strain was calculated by the following formula: BE (%) = (weight of fresh stroma/weight of dry substrate) × 100 [33,34]. The biological efficiency, number and height of fruiting bodies of each vessel were measured in three groups (5 per group). These cultivation experiments were carried out three times independently.

### 2.9. Expression of CmAreA in E. coli and Electrophoretic Mobility Shift Assay (EMSA)

The cDNA sequence of *Cmarea* was cloned into vector PET28a (provided by Prof. Linqi Wang from the Institute of Microbiology, Chinese Academy of Sciences) with a 6×His-tag and introduced into *E. coli* BL21-DE3 (Biomed Biotech Co., Ltd., Beijing, China). Expressed CmAreA was purified using nickel–nitrilotriacetic acid (Ni-NTA, Sangon Biotech Co., Ltd., Shanghai, China).

A 50 bp DNA probe containing the GATA sequences from the *Cmhyd1* promoter was labeled with digoxigenin-11-ddUTP (DIG-11-ddUTP, Beyotime Biotech Co., Ltd., Beijing, China) at the 3′ end (Appendix A). EMSA was performed with a Chemiluminescent EMSA Kit (Beyotime Biotech Co., Ltd., Beijing, China), following the manual.

### 2.10. Transient Luciferase Expression in Tobacco

The 0.36 kb promoter of *Cmhyd1* was cloned into the vector pGWB435 (provided by Prof. Dapeng Zhang from Tsinghua University) with a *LUC* reporter gene to generate P*_Cmhyd1_*: LUC as a reporter. The coding sequences of *Cmarea* and *Cmotam* were cloned into vector pBI121 (provided by Prof. Naiqin Zhong from the Institute of Microbiology, Chinese Academy of Sciences) to generate P*_35S_*: CmAreA and P*_35S_*: CmOTam as effectors. All the vectors were transformed to *A. tumefaciens* strain GV3101 (Zoman Biotech Co., Ltd., Beijing). The strains were incubated in YEB media and finally resuspended in infiltration buffer (10 mM MES, 10 mM MgCl_2_ and 0.15 mM AS) to an ultimate concentration with optical density of 600 nm = 1. Equal amounts of different combined bacterial suspensions were infiltrated into the young leaves of 5-week-old tobacco (*Nicotiana tabacum*) plants using a needleless syringe. Two days later, the infected leaves were sprayed with 100 μM luciferin (Sigma-Aldrich, Darmstadt, Germany) and placed in the dark for 3 min [35]. The LUC signal was detected using the Fusion FX system (Vilber Lourmat, Paris, France). Fluorescence intensity was represented by OD_450_ with the ELISA instrument Infinite F50 (Tecan, Switzerland, Austria). Experiments were performed with ten independent biological replicates.

### 2.11. RNA Extraction and RT-qPCR Analysis

Total RNA was extracted from frozen mycelia or fruiting bodies using an E.Z.N.A.™ Plant RNA Kit (Omega, Stamford, CT, USA) according to the manufacturer’s protocol. First-strand cDNA was synthesized from 1 μg total RNA using a HiScript III 1st Strand cDNA Synthesis Kit (+gDNA wiper) (Vazyme Biotech Co., Ltd., Beijing, China). Quantitative real-time PCR (qPCR) was conducted using AceQ^®^ qPCR SYBR^®^ Green Master Mix (Q111) qPCR mix (Vazyme Biotech Co., Ltd., Beijing, China) and a CFX Connect Real-Time System (Bio-Rad, Singapore). Relative gene expression levels were calculated using the 2^−ΔΔCt^ method (Livak and Schmittgen, 2001). The *rpb1* (CCM_05485) gene was used as an internal standard [36]. All the primer sequences used in this study are listed in Appendix A. The obtained data represented three biological replicates, with two technical replicates each.

## 3. Results

### 3.1. Deletion, Complementation and Overexpression of Cmhyd1 in Cordyceps militaris

*Cmhyd1* knockouts (Δ*Cmhyd1*), genetic complementation (Δ*Cmhyd1c*) and overexpression strains (*Cmhyd1oe*) were generated in *C. militaris* (Appendix A). Δ*Cmhyd1* and Δ*Cmhyd1c* strains were identified by PCR with paired primers (Appendix A), then confirmed by RT-PCR and Southern blot hybridization with an amplified probe. The RT-PCR results revealed that *Cmhyd1* of the knockout strains was transcriptionally inactive (Appendix A). The wild type (WT) and Δ*Cmhyd1* strains showed hybridized bands of 1.09 kb and 3.03 kb, respectively, whereas Δ*Cmhyd1c* strains displayed hybridized bands of both 1.09 kb and 3.03 kb (Appendix A). The overexpression strains were confirmed by RT-PCR and the expression levels of *Cmhyd1* were about 1.6–1.8-fold that of the WT strain (Appendix A). One Δ*Cmhyd1* strain, two complementation strains (Δ*Cmhyd1c-1* and Δ*Cmhyd1c-3*) and two overexpression strains (*Cmhyd1oe-2* and *Cmhyd1oe-4*) were obtained and used for the phenotype observation.

### 3.2. Deletion of Cmhyd1 Affects Cordyceps militaris Morphology and Hydrophobicity

The aerial mycelia of the Δ*Cmhyd1* strain were thinner than that of WT and *Cmhyd1oe* strains, as shown in Figure 1A. The contact angle of droplets on the surface of mycelia was used to define the hydrophobicity of the mycelia and a large contact angle indicates strong hydrophobicity [37]. Obviously, the aerial mycelia of the Δ*Cmhyd1* strain showed reduced hydrophobicity (Figure 1B). The growth rate of the Δ*Cmhyd1* strain was significantly lower than that of the WT strain on PDA medium (Figure 1D,E). The observation by scanning electron microscopy (SEM) showed that the hyphae of the WT strain were smoother than those of the Δ*Cmhyd1* strain, which were usually enwound and adhered to each other (Figure 2A,B). Obvious differences in colony color were observed when all the strains were cultured on PDA plates after being irradiated for 4 d. The colony of the Δ*Cmhyd1* strain remained white. The mycelium color of the *ΔCmhyd1c* culture was similar to that of the WT, while the color of the *Cmhyd1oe* culture was much darker than that of the WT (Figure 1D). These observations suggested that *Cmhyd1* could be related to pigment synthesis in *C. militaris*.

Conidiation of the Δ*Cmhyd1* strain was significantly suppressed compared with that of the WT strain. The two complementation strains Δ*Cmhyd1c-1* and Δ*Cmhyd1c-3* showed a similar conidial yield as the WT strain while there was a significant increase in the Δ*Cmhyd1oe-2* and Δ*Cmhyd1oe-4* strains compared to the WT strain (Figure 1F). Meanwhile, the conidia of Δ*Cmhyd1* were prone to grouping together, indicating that the hydrophobicity of the Δ*Cmhyd1* conidia was reduced (Figure 1C). The results were also confirmed by the MATH assay: Triton X-100 at 1.0 mL/L was sufficient to release all the conidia of the Δ*Cmhyd1* strain whereas 6.0 mL/L was needed for the WT strain (Appendix A). There were many more granular attachments on the surface of the conidia of Δ*Cmhyd1* than that of the WT, observed by SEM (Figure 2A,B).

### 3.3. Deletion of Cmhyd1 Affects Stress Responses in Cordyceps militaris

In comparison with the WT and complementation strains, the Δ*Cmhyd1* strain grew more slowly in the presence of 10% CR, 0.1% SDS and 0.075 mM H_2_O_2_ while the overexpression strains grew more quickly (Figure 3A). It was indicated that cell wall integrity and tolerance to oxidant stress were affected in the Δ*Cmhyd1* strain.

Germination rates of conidia from all the strains were determined after high (35 °C, 40 °C and 45 °C) or low temperature (−80 °C, −20 °C and 0 °C) treatments for 0–3 h. The results showed that the conidia of all the strains could germinate except those treated at a higher temperature of 45 °C for over 1 h. Germination rates of conidia of all the strains decreased with the increase in stress time under both high and low temperatures (Figure 3B). Among all the strains, the Δ*Cmhyd1* strain had the lowest germination rate and *Cmhyd1oe* strains had the highest under all the treatments (Figure 3B).

### 3.4. Cmhyd1 Is Associated with Infection in Silkworm Pupae

The conidia of *C. militaris* can infect silkworm pupae and thus form club-like fruiting bodies in the field or by artificial cultivation. The Δ*Cmhyd1* strains exhibited the lowest cuticle-bypassing infection rates during the whole period among all the tested strains and it took the longest time for the infected pupae to be mummified completely (Figure 4A), implying that *Cmhyd1* played an important role in cuticle-bypassing infection.

CmHYD1 was expressed in *Pichia* and the purified CmHYD1 protein was added to the conidia of Δ*Cmhyd1* to determine the infection ability. It was found that the virulence was increased and close to the cuticle-bypassing infection ability of WT and Δ*Cmhyd1c* strains (Figure 4A,B). The infection rates of WT with exogenous CmHYD1 were actually close to those of *Cmhyd1oe* strains (Figure 4A,B).

### 3.5. Deletion of Cmhyd1 Affects the Transcript Level of Conidiation and Infection-Related Genes during Conidium Production and Infection

Genes encoding for frequency clock protein (*frq*) [38]*,* velvet protein (*vosa*) [39] and a transcription factor (*flug*) [40] were proved to play roles in conidium development and infection in *B. bassiana*. The transcript levels of the homologous genes in *C. militaris*, *Cmfrq* (CCM_01014), *Cmvosa* (CCM_02384) and *Cmflug* (CCM_08119), were evaluated in the WT and Δ*Cmhyd1* strains under darkness and light irradiation for 2 d since light is necessary for conidiation [41]. The transcript level of *Cmhyd1* was up-regulated 3-fold after being irradiated for 2 d compared with darkness in the WT strain (Figure 5A). Transcripts of *Cmfrq* and *Cmvosa* were up-regulated over 4-fold after being irradiated for 2 d compared with darkness in the WT strain while very low up-regulation occurred in the Δ*Cmhyd1* strain (Figure 5A and Appendix A). The transcript level of *Cmflug* was significantly down-regulated after being irradiated for 2 d in both WT and Δ*Cmhyd1* strains (Figure 5A and Appendix A). Transcripts of *Cmfrq* and *Cmvosa* were down-regulated in Δ*Cmhyd1* compared with the WT strain under both darkness and light conditions (Figure 5A), which can explain the significant decrease in conidium production in the Δ*Cmhyd1* strain.

The pupae of silkworm are mummified after a cuticle-bypassing infection of about 16 d by *C. militaris*. Since the conidia were used to infect the pupae, the transcript levels of *Cmhyd1* and conidiation-related genes, *Cmfrq*, *Cmvosa* and *Cmflug*, were also evaluated during the infection process, including the stages of mycelia (MY) and infection for 8 d (STE) and 16 d (STL). The mRNA levels of *Cmhyd1* rose rapidly at both early and late infection stages compared with the MY stage. The transcript levels of *Cmfrq*, *Cmvosa* and *Cmflug* increased during the infection compared with the MY stage and were higher at the STE than STL stage in both the WT and Δ*Cmhyd1* strains, however, the increase was much lower in the Δ*Cmhyd1* strain (Figure 5B and Appendix A).

### 3.6. Cmhyd1 Is Important in Fruiting Body Development in Cordyceps militaris

As one of the well-known edible and medicinal mushrooms, fruiting body development is very important for *C. militaris*. The Δ*Cmhyd1* strain cannot form primordium and a fruiting body, while the fruiting bodies of Δ*Cmhyd1c* and *Cmhyd1oe* strains can develop well (Figure 6A). The WT strain with exogenous CmHYD1 produced much more primordia and had the highest biological efficiency, which was significantly higher than that of the WT strain (Figure 6A,B). The results confirmed our previous conclusion that *Cmhyd1* plays a very important role in primordium differentiation and fruiting body development, revealed by the transcript profile [23]. The fruiting body of Δ*Cmhyd1c* strain could develop well, however, there was no primordium differentiation in the Δ*Cmhyd1* strain with exogenous CmHYD1 (Figure 6A,B).

The transcript levels of *Cmhyd1* and fruiting body development-related genes were compared between the stages of mycelia and primordia. Genes in the MAPK signal pathway, *Pth11* (Pth11-like G-protein coupled receptor*,* CCM_03015), *CDC42* (Ras small GTPase, CCM_00979), *STE7* (MAPKK, CCM_03428), *STE11* (MAPKKK, CCM_02296), *STE20* (MAPKKKK, CCM_09268) and *ERK* (mitogen-activated protein kinase, CCM_01235, CCM_09637) [29] were tested. At the primordium stage, the transcript level of *Cmhyd1* was increased over 40-fold and 20-fold in the WT strain and WT strain with exogenous CmHYD1, respectively (Figure 6C). All the tested genes related to fruiting body development except *ERK1* and *ERK2* showed significant up-regulation at the primordium stage compared with the mycelial stage, which confirmed that the MAPK signal pathway was responsible for the fruiting body development in *C. militaris*. This up-regulation at the primordium stage disappeared for the genes *Pth11*, *CDC42*, *STE7*, *STE11* and *STE20* in Δ*Cmhyd1* strains. Transcript levels of *Pth11*, *CDC42* and *STE20* were up-regulated in the Δ*Cmhyd1* strain with exogenous CmHYD1 (Figure 6C and Appendix A). 

### 3.7. Transcript of Cmhyd1 Is Regulated by CmAreA Cooperating with CmOtam

CmAreA contains a GATA zinc finger (ZnF) binding domain at the region spanning amino acids (aa) 609 to 658 of its C terminus (Appendix A), which may recognize and bind a core (A/T)GATA(A/G) consensus sequence [42]. Phylogenetic analysis showed that AreA is a conserved GATA transcript factor in fungi (Appendix A). 

The transcript levels of *Cmhyd1* and *Cmarea* (CCM_03599) were spatially and temporally coincident during all developmental stages of *C. militaris*, including mycelium (MY), sclerotium (ST), young primordium (YPR), primordium (PR), young fruiting body (YF), down part of developed fruiting body (DF) (DF-D) and up part of DF (DF-U), down sterility stipe of mature fruiting body (MF) (MF-D) and up fertile part of MF (MF-U), as in our previous description [23] (Figure 7A). Both *Cmhyd1* and *Cmarea* showed higher transcription levels from ST to MF stages than the MY stage (Figure 7A). As AreA is a core transcription factor of the nitrogen metabolism pathway in some filamentous fungi [43,44], the transcript levels of *Cmhyd1* and *Cmarea* at the primordium stage with different ammonium citrate concentrations were determined. The results showed the same synchronized transcript between *Cmhyd1* and *Cmarea*, with the highest transcription levels at a concentration of 1.0 g/L ammonium citrate (Figure 7B).

Scanning of a 364 bp *Cmhyd1* promoter region identified two core GATA sequences located at nucleotides −81 and −14 (Figure 7C). To confirm whether CmAreA, as a GATA transcription factor, could directly bind to the GATA sequences in the *Cmhyd1* promoter region, an in vitro electrophoretic mobility shift assay (EMSA) with two probes was performed. Each of probe contained one predicted GATA consensus sequence, as shown in Figure 7C. CmAreA increasingly bound to DNA probe 1 with increasing recruitment of CmAreA protein (Figure 7D). In the full EMSA, CmAreA efficiently bound to GATA-box DNA biotin probe 1 (Figure 7E, lane 2). The specificity of the CmAreA–DNA complex was confirmed by adding an excess of unlabeled DNA (Figure 7E, lane 3). As expected, a shift was detected when adding the unlabeled DNA and GATA-box DNA probe 1 (Figure 7E, lane 4). The same shifts were also checked in the EMSA of GATA-box DNA probe 2 (Figure 7F). These data suggested that CmAreA was recruited to the two GATA boxes in the *Cmhyd1* promoter in vitro.

Then, the results were confirmed by an in vivo transient luciferase expression system in tobacco. The *Cmhyd1* promoter was fused with a luciferase (LUC) reporter gene to generate P*_Cmhyd1_*: LUC as a reporter, and the coding sequence of *Cmarea* was cloned into vector pBI121 to generate P*_35S_*: CmAreA. The resulting P*_Cmhyd1_*: LUC and P*_35S_*: CmAreA were cotransformed into tobacco leaves. A weak luminescence signal was observed, indicating that CmAreA could activate the *Cmhyd1* promoter although this activity was not strong (Figure 7G). Small et al. [45] found that the *tamA* gene, another GATA ZnF transcriptor factor, has a positive role together with *areA* in regulating gene expression in *A. nidulans*. The TamA homologous protein, CmOtam (CCM_04044) in *C. militaris*, was coexpressed with CmAreA protein in tobacco and it was found that there was a stronger activation, while only CmOTam could not activate the LUC reporter (Figure 7G). These results indicated that CmAreA can activate the transcription of *Cmhyd1* with the coactivator of CmOTam.

Furthermore, the *Cmarea* gene was disrupted (Appendix A) and the development-associated phenotypes of Δ*Cmarea* strains were compared with WT and Δ*Cmhyd1* strains. Both the Δ*Cmhyd1* and Δ*Cmarea* strains had thinner aerial hyphae, produced much fewer conidia after light treatment on PDA media than the WT strain, showed low hydrophobicity for mycelia and conidia and remained white and could not form a primordium and fruiting body after light irradiation, which was totally different from WT strains (Figure 1). The same hypha adhesion and enwinding with the Δ*Cmhyd1* strain were abserved in Δ*Cmarea* strains by SEM (Figure 2C). With regard to pathogenicity, the healthy silkworm pupae were mummified completely by the WT strain after inoculation for 19 d, while it took 27 d and 24 d, respectively, for the Δ*Cmarea* and Δ*Cmhyd1* strains (Figure 4A and Figure 8A). The time until full mummification of healthy pupae inoculated by the Δ*Cmarea* strain was reduced to 21 d after adding exgenous hydrophobin CmHYD1, however, the primordium could not yet form (Figure 8A,B). It seemed that most development-associated phenotypes were shared by Δ*Cmhyd1* and Δ*Cmarea* strains. 

The transcript levels of *Cmhyd1* were up-regulated during conidiation and infection, as well as primordium formation in the WT strain, but reduced drastically in the Δ*Cmarea* strain (Figure 8C). This suggested that the transcription regulator CmAreA is necessary for the transcription of *Cmhyd1*.

### 3.8. CmHYD1 Positively Regulates the Expression of the Transcription Factor Cmarea

The deletion of *Cmarea* impaired the infection ability significantly. Amazingly, the time to becoming mummified completely was shortened by 6 d in the Δ*Cmarea* strain with exgenous CmHYD1, although it still could not form a primordium (Figure 8A,B). Then, the *Cmarea* and *Cmhyd1* genes under the control of their original promoters and terminators were expressed in the Δ*Cmarea* strain. It was found that *Cmhyd1* could not restore the ability of pigment synthesis, infection and primordium differentiation, unlike with *Cmarea* (Figure 8A,B).

To futher explore the interaction relationship between *Cmhyd1* and *Cmarea*, the transcript levels of the *Cmarea* gene in the Δ*Cmhyd1* strain were determined. It was found that the transcript levels of *Cmarea* were up-regulated 12-fold in the WT strain while only 3-fold in the Δ*Cmhyd1* strain during conidiation (Figure 8D). During infection, the transcript level of *Cmarea* was drastically increased about 10-fold, while it remained stable in the Δ*Cmhyd1* strain compared with the mycelia of the WT (Figure 8D). The transcript level of *Cmarea* increased significantly when exgenous CmHYD1 was added to the conidia of WT and Δ*Cmhyd1* strains during the early stage of infection (STE). During the primordium formation, the transcript level of *Cmarea* was down-regulated in the Δ*Cmhyd1* strain but up-regulated 30-fold in the Δ*Cmhyd1* strain after adding exgenous CmHYD1. This suggested that CmHYD1 positively regulated the transcript levels of *Cmarea* during conidiation, infection and primordium formation.

## 4. Discussion

Four hydrophobin-encoding genes have been reported in this fungus and *Cmhyd1* showed a high transcript level during fruiting body development compared with the mycelial stage [23]. In this study, it was found that *Cmhyd1* was involved in aerial mycelial development, conidiation, hydrophobicity of mycelia and conidia, pigment synthesis, cuticle-bypassing infection and primordium formation by gene deletion, overexpression and complementation. Furthermore, it was found that the transcription of *Cmhyd1* was activated by the transcription factor CmAreA using the EMSA in vitro, transient luciferase expression in tobacco and *Cmarea* gene disruption in *C. militaris*. On the other hand, CmHYD1 regulated the transcription of *Cmarea* at different developmental stages via a positive feedback loop. The understanding of the function and regulatory mechanism of hydrophobins in *C. militaris*, a well-known edible and medicinal mushroom, will undoubtedly be helpful for research on the development of multicellular fungi.

Similar with the other entomopathogenic fungi, *B. bassiana* and *Me. brunneum*, deletion of *Cmhyd1* resulted in a decreased mycelial growth rate, conidiation, hydrophobicity and fungal virulence. Deletion of *Cmhyd1* in *C. militaris* suppressed sporulation, but did not eliminate it entirely, which was consistent with *T. reesei* and *Me. brunneum* [9,46]. It was suggested that the function of this hydrophobin gene may be partially compensated by the other three hydrophobins. Deletion of *Cmhyd1* decreased the hydrophobicity of both conidia and hyphae, which resulted in the adhesion of some hyphae and conidia (Figure 1B,C and Figure 2A,B). Meanwhile, it was reported that deletion of *Hyd1* did not influence mycelial hydrophobicity in *Me. brunneum* or *C. rosea* [11]. Ball et al. [47] showed that rodlets were observed on the outer surface of the aerial hyphae and conidia in the cell walls in some fungi, and hydrophobins were the proteins that made up the rodlet layer. For *B. bassiana*, it was indicated that HYD1 and HYD2 were the structural components of the rodlet layer [8]. However, no rodlets have been observed on the outer surface of conidia for either the WT or Δ*Cmhyd1* strains in *C. militaris*. The localization of CmHYD1 in *C. militaris* remains to be further studied. There was increased ornamentation on the surface of the conidia in the Δ*Cmhyd1* strain (Figure 2B). Deletion of hydrophobin genes in different fungal species often results in variable and sometimes contradicting phenotypes. This is a reflection of the birth and death type of evolution of the hydrophobin gene family [48], which results in functionally diverse proteins.

After being exposed to light, the colony color of *C. militaris* transforms from white to yellow or orange, and then the primordia begin to differentiate and develop [49]. It was unexpected that the Δ*Cmhyd1* strain maintained the white phenotype after light irradiation (Figure 1D) since our previous study revealed that there was almost no light response in the transcription of *Cmhyd1* in both the WT and photoreceptor mutants Δ*Cmwc-1* and Δ*Cmvvd* [23]. There are also alterations in pigmentation in the hydrophobin gene deletion strains of *Me. brunneum* [9]. This suggested that *Cmhyd1* may regulate the accumulation of the pigment in *C. militaris*.

*Cmhyd1* also participated in stress response in *C. militaris*, including the tolerance of mycelia to osmotic and oxidative stresses, and conidia to high or low temperature. Since the cell wall is a physiological barrier against environmental stress [50], hydrophobin proteins form a hydrophobic layer that coats the cell wall polysaccharides and provides resistance to external stresses [47,51,52]. It was assumed that the mycelium- and conidium-related phenotypic changes in the Δ*Cmhyd1* strain (Figure 1A–D and Figure 2B) may have resulted from the changes in the cell wall properties. The fact that overexpression of *Cmhyd1* increased the stress tolerance confirmed the results. There were completely different results for *B. bassiana*, in which hydrophobins did not appear to significantly contribute to protection from a range of cell wall stressors, including calcofluor white, Congo red and high salt, but the presence of the *Hyd1* hydrophobin appeared to destabilize the thermal resistance of the spores [8]. It was suggested that there may be a potential trade-off among the benefits of the hydrophobins with respect to spore stability, adhesion or other phenotypes during the evolution of hydrophobins. In *Aspergillus sydowii* (Eurotiales), a halophilic species, three hydrophobin genes were only differentially expressed under hypo- or hyper-osmotic stress but not when the fungus grew optimally [53]. Protection against oxygen stress by *Cmhyd1* was also consistent with a previous demonstration of antioxidant activities of *HFB2*, a hydrophobin in *T. reesei* [54].

Studies in pathogenic fungi led to the suggestion that hydrophobins may be important in fungal–host interactions [8,55]. Deletion of *Cmhyd1* resulted in the delaying of the mummification of its host (Figure 4), and adding exogenous CmHYD1 accelerated cuticle-bypassing infection in both WT and Δ*Cmhyd1* strains in *C. militaris* (Figure 4). This implied that CmHYD1 played a key role in the virulence to the insects. As reported for entomogenous fungi, the hydrophobin family shows a remarkable function in infection with fungi [4,7,8,9]. The transcription of conidiation-related genes *Cmfrq* and *Cmvosa* was down-regulated during conidiation and infection in response to *Cmhyd1* disruption (Figure 5B). 

In general, hydrophobins are thought to regulate fruiting body development in mushroom-forming fungi. However, direct evidence was only found in *S. commune* [56]. Recently, the role in the formation of primordia was verified in *F. filiformis* and *P. ostreatus* by RNAi [16,17]. In this study, it was confirmed that *Cmhyd1* was necessary for primordium formation in *C. militaris* by gene deletion, complementation, overexpression and exogenous addition of CmHYD1. Zheng et al. [29] reported that fruiting by *C. militaris* is more dependent on the MAPK pathway than the cAMP-dependent PKA pathway, which was confirmed by the up-regulation of the transcription of genes in the MAPK pathway during primordium formation in this study. Meanwhile, those genes in the MAPK pathway were down-regulated during primordium formation in the *Cmhyd1* deletion strain. On the other hand, there was no significant differences in the biological efficiencies between the *Cmhyd1* overexpression strain and WT strain with exogenous CmHYD1 (Figure 6), but both were significantly higher than that of the WT strain. This suggested that CmHYD1 protein may function as a signal to promote fruiting body development. 

Hydrophobin gene MPG1 in *Ma. grisea* is regulated by nitrogen repression genes (NPR1 and NUT1) [25] and hydrophobin gene *Mrhyd4* in *Me. robertsii* is regulated by a carbon catabolite repressor, MrCre1 [27,28]. It was suggested that a nutrition-linked regulatory mechanism may be involved in the regulation of hydrophobin genes. In this study, a core transcription factor of nitrogen metabolism, CmAreA, was confirmed to directly bind to the promoter of *Cmhyd1* by the EMSA method in vitro (Figure 7C–F). In the experiments of transient luciferase expression in tobacco, there was a stronger activation when CmAreA was coexpressed with CmOtam, while only CmOTam or CmAreA could not activate the LUC reporter or only weakly (Figure 7G). This suggested that CmAreA can activate the transcription of *Cmhyd1* with the coactivator, CmOTam. It has also been reported that the *tamA* gene has a positive role together with *areA* in regulating gene expression in *A. nidulans* [45].

Phenotypically, both the Δ*Cmhyd1* and Δ*Cmarea* strains exhibited similar defective development with respect to the aerial hypha, hydrophobicity, photoreaction, conidiation, infection and primordium differiation (Figure 1, Figure 2, and Figure 8A,B). The transcript levels of *Cmhyd1* during conidiation, infection and primordium formation were extremely low in the Δ*Cmarea* strain. It was confirmed that CmAreA could activate the transcription of *Cmhyd1* in *C. militaris*, which is the first report about the direct connection between the transcript factor AreA and hydrophobin genes in edible fungi.

As well as CmAreA activating the transcription of *Cmhyd1*, it was also found that CmHYD1 can positively regulate the transcript levels of *Cmarea* during conidiation, infection and primordium formation by monitoring the gene transcript profile and by adding exgenous CmHYD1 (Figure 8D). The transcript levels of *Cmarea* were dependent on CmHYD1 since the up-regulation was much more suppressed in the Δ*Cmhyd1* strains compared with WT (Figure 8D) and these suppressions can be restored partially when adding exgenous CmHYD1 (Figure 8D). The infection ability was also improved in the Δ*Cmarea* strain by adding exgenous CmHYD1 (Figure 8A), confirming the important role of *Cmhyd1* in the virulence. These data suggested that CmHYD1 up-regulated the transcription of *Cmarea* at different development stages via a positive feedback loop, which eventually magnified senescence signals and resulted in the formation of hydrophobic structures in *C. militaris*. Therefore, the positive transcriptional feedback mechanism mediated by nitrogen regulatory GATA transcript factor CmAreA, described here, was crucial for the development of hydrophobic structures and could be, at least in part, responsible for the unique dynamics of the nitrogen regulatory pathway output. This positive feedback mechanism is similar to SLT2- and RLM1-mediated positive feedback mechanisms in the cell wall integrity (CWI) pathway of *Saccharomyces cerevisiae* [57].

Altogether, our findings unveiled diverse roles for *Cmhyd1* in hypha, conidiation, pigmentation, infection and primordium formation of *C. militaris. Cmhyd1* played roles in conidiation and infection by regulating the transcript levels of *Cmfrq* and *Cmvosa*, and primordium formation through the MAPK signaling pathway. CmAreA was proved to directly bind to the promoter of *Cmhyd1* and activate the transcription of *Cmhyd1* with coactivator CmOTam. Furthermore, it was found that CmHYD1 can regulate the transcript level of *Cmarea* via a positive feedback mechanism (Figure 9).

## Figures and Tables

**Figure 1 jof-07-00674-f001:**
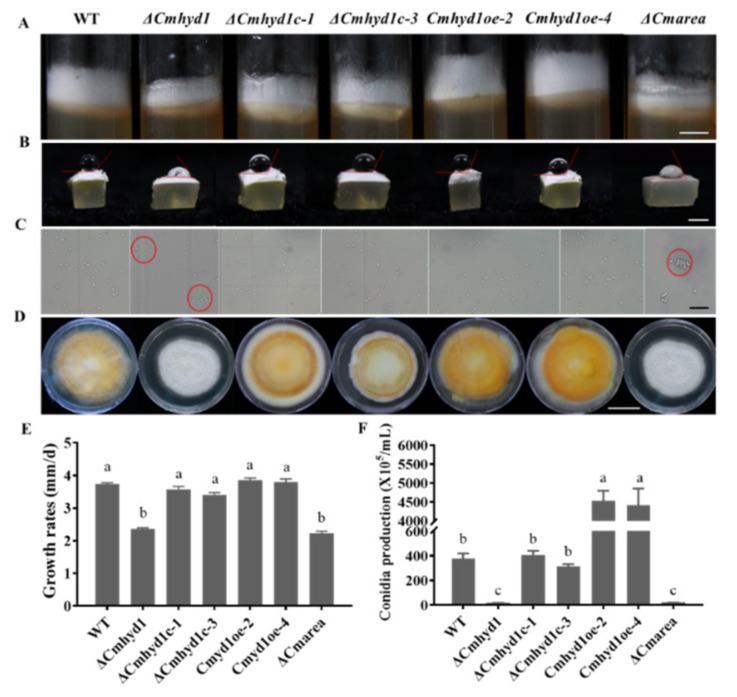
Morphology, growth rate and conidiation in the WT and mutants. (**A**) Aerial mycelia. Aerial mycelia were observed after being cultured for 7 d at 20 °C in a test tube with PDA medium. Bar: 1 cm. (**B**) The hydrophobicity of mycelia. Bar: 1 cm. (**C**) The hydrophobicity of conidia. Bar: 20 μm. Red circles indicate the conidia which were prone to grouping together. (**D**) Colony morphology, observed after incubation for 21 d at 20 °C on PDA medium and then exposed to 12/12 h white light/dark conditions for 4 d. Bar: 3 cm. (**E**) Growth rates. (**F**) Conidium production. Error bars indicate the standard deviation (SD) of three replicates. Different letters above the bars indicate significant differences (analysis of variance followed by Duncan’s multiple range test, *p* < 0.05).

**Figure 2 jof-07-00674-f002:**
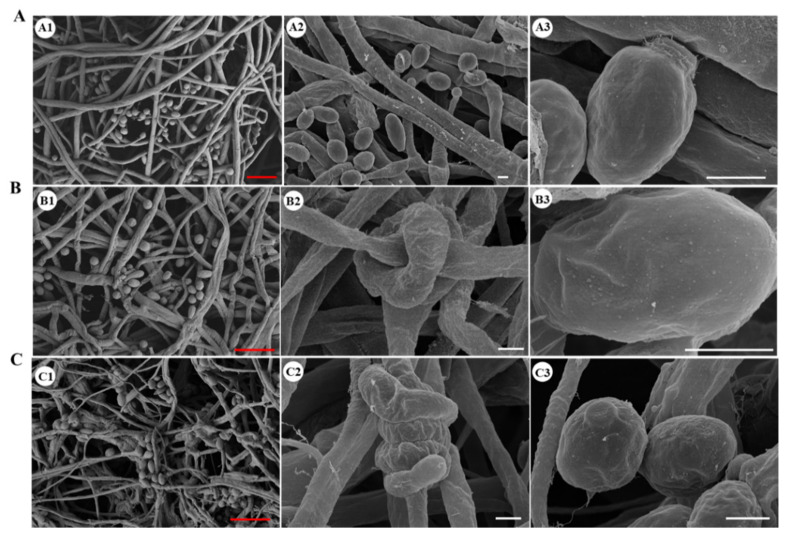
Conidia and hyphae of WT and mutant strains observed by SEM. (**A**) WT strain. (**B**) Δ*Cmhyd1* strain. (**C**) Δ*Cmarea* strain. Bars: A1, B1, C1 = 1 μm; others = 10 μm.

**Figure 3 jof-07-00674-f003:**
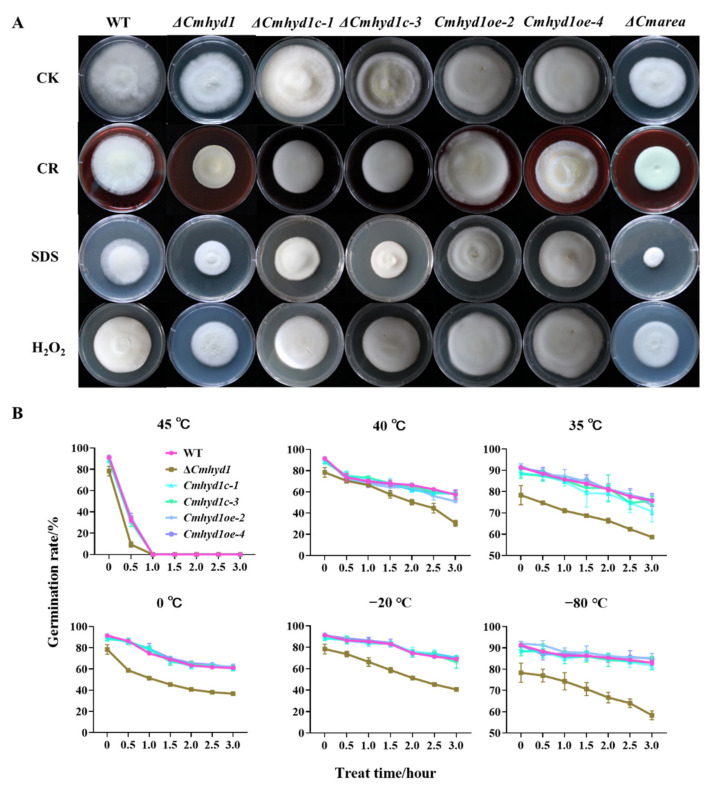
The stress response of mycelia and conidia in the WT and mutants. (**A**) Sensitivity of WT and all mutant strains against various stresses. CK was the control inoculated on PDA medium. CR, SDS and H_2_O_2_ indicated that the strains were inoculated on PDA with 10% Congo red, 0.1% SDS and 0.075 mM H_2_O_2_. All the strains were cultured at 20 °C under darkness for 21 d and photographed. (**B**) Germination rates of the conidia under high or low temperatures. Conidia were treated for 0 to 3 h and then cultured in 1 mL PDB medium. Error bars indicate the standard deviation (SD) of three replicates.

**Figure 4 jof-07-00674-f004:**
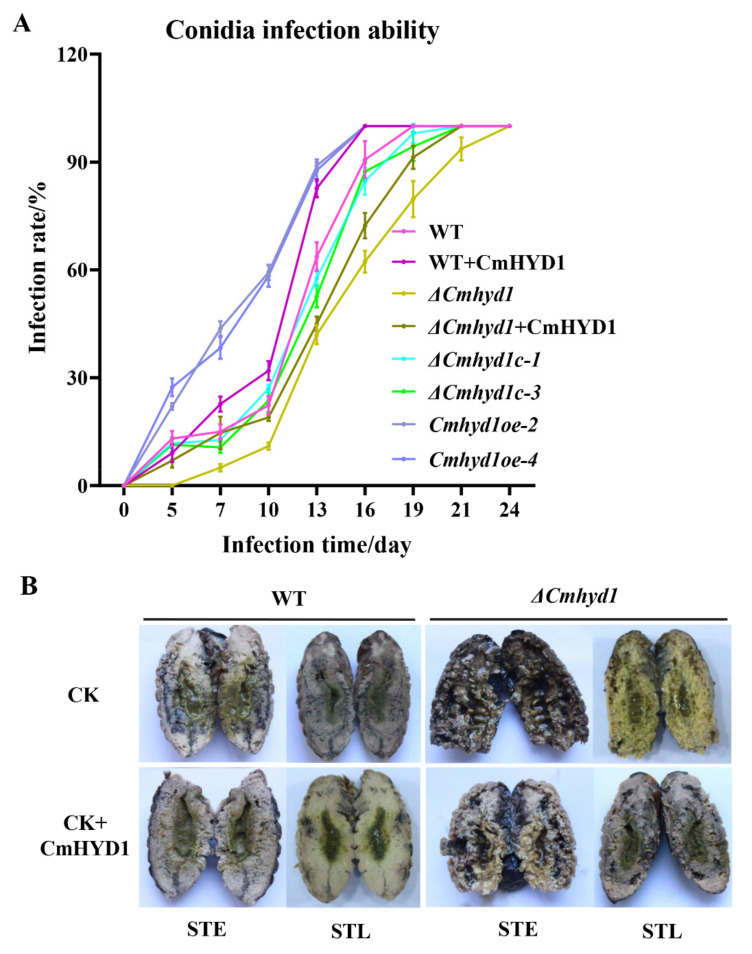
Cuticle-bypassing infecting ability of conidia of WT and mutants. (**A**) Infection rates of conidia and conidia with exogenous CmHYD1. (**B**) Longitudinal sections of silkworm pupae after being infected for 8 d (STE, early sclerotia) and 16 d (STL, late sclerotia) by conidia of WT, Δ*Cmhyd1* and addition of exogenous CmHYD1.

**Figure 5 jof-07-00674-f005:**
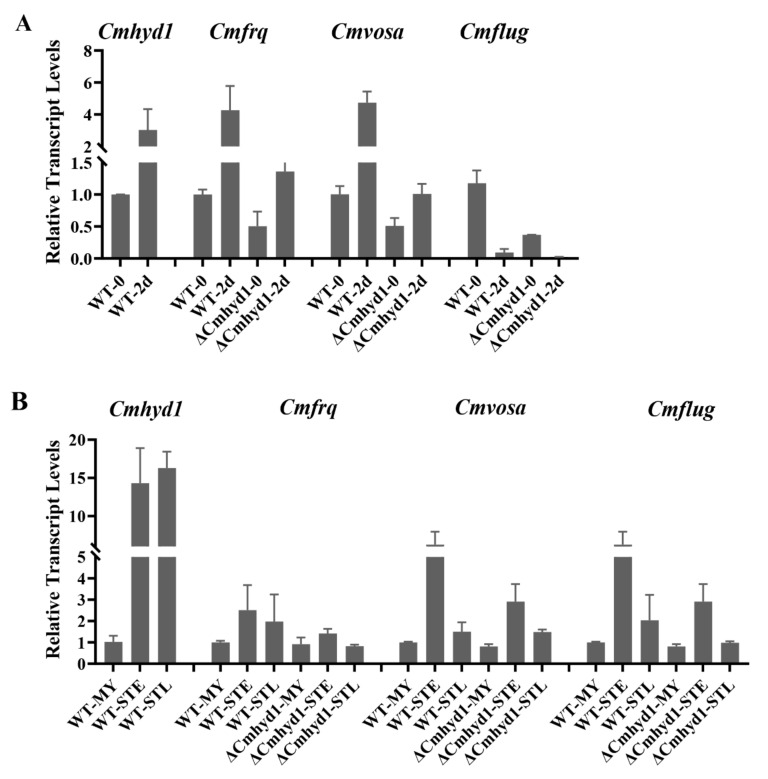
Transcript analysis of *Cmhyd1* and conidiation-related genes in Δ*Cmhyd1* and WT strains during conidiation and infection. (**A**) Transcript levels of *Cmhyd1* and conidiation-related genes after being exposed to light for 2 d. All results were based on the standard levels of darkness in WT strain. WT-0 and ΔCmhyd1-0 indicated the WT and Δ*Cmhyd1* strains cultured in darkness, WT-2 and ΔCmhyd1-2 indicate they were exposed to light for 2 d. (**B**) Transcript levels of *Cmhyd1* and conidiation-related genes at different infection stages. All results were based on standard levels of mycelial stage in WT strain. MY, STE and STL indicate the stages of mycelia, early sclerotia and late sclerotia, as shown in Figure 4B. Error bars indicate the standard deviation (SD) of three biological replicates with two technical replicates.

**Figure 6 jof-07-00674-f006:**
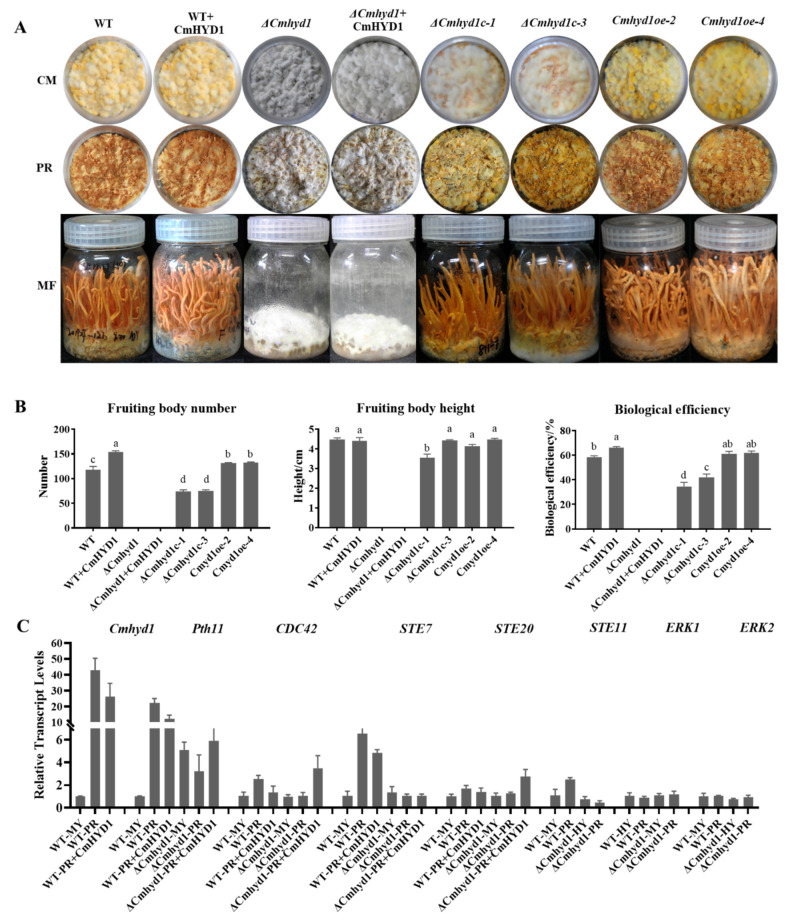
*Cmhyd1* regulates fruiting body development in *Cordyceps militaris*. (**A**) Fruiting body development of WT and mutants. CM: the colored mycelia after light irradiation for 4 d; PR: primordium formation (stroma < 1 cm); MF: mature fruiting body. “WT + CmHYD1” and “Δ*Cmhyd1* + CmHYD1” indicate that exogenous CmHYD1 was added to the WT and Δ*Cmhyd1* strains cultured on wheat media. (**B**) Effects of *Cmhyd1* on mushroom yield. Fruiting bodies were harvested on 45th day after inoculation. The different letters over the histogram indicate significant differences at *p* < 0.05. (**C**) The relative transcript levels of *Cmhyd1* and fruiting body development-related genes at primordial stage. All the results were based on the standard levels of the mycelial stage (MY) of WT strain. Error bars indicate the standard deviation (SD) of three biological replicates with two technical replicates.

**Figure 7 jof-07-00674-f007:**
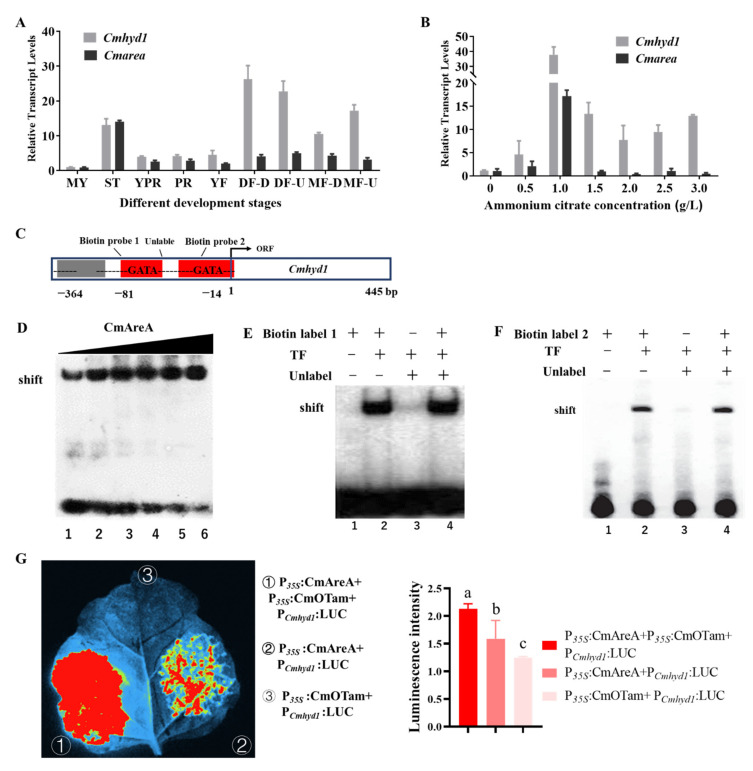
The transcription factor CmAreA was recruited to the promoter and activated the transcript of of *Cmhyd1*. (**A**) The relative transcript levels of *Cmhyd1* and *Cmarea* at different developmental stages. MY: mycelium used for inoculation of silkworm pupae; ST: sclerotium (mummified silkworm pupae) before stroma development; YPR: mycelium knot tissue before stroma development; PR: sclerotium with initial stroma (stroma < 1 cm); YF: sclerotium with early stage stroma (1 cm < stroma < 2 cm); DF: sclerotium with developed stroma (stroma > 5 cm); MF: mature fruiting body. The results were based on the standard levels of the MY stage. (**B**) The relative transcript levels of *Cmhyd1* and *Cmarea* at primordial stage of WT with different concentrations of ammonium citrate. (**C**) The schematic diagram showing the predicted GATA-binding elements in the promoter region of *Cmhyd1*. The transcription start sites are indicated as “1”. ORF: open reading frame. Biotin probe: the biotin probe for EMSA. Unlabel: negative control for EMSA. (**D**) Characterization of CmAreA interaction with GATA-binding site by EMSA. A 50 bp *Cmhyd1* promoter probe was incubated with a quantity of CmAreA protein (400 ng, 600 ng, 800 ng, 1000 ng, 1500 ng or 3000 ng). With the increase in the amount of CmAreA, a greater DNA shift was observed with less free probe accumulation at the bottom. (**E**,**F**) CmAreA can bind to the two GATA sequences in the *Cmhyd1* promoter by EMSA. (**E**,**F**) The results of probe 1 and 2, respectively. The labeled DNA probe 1 or 2 was preincubated with 800 ng CmAreA, and an unlabeled competitor was added. The presence or absence of the reaction is indicated with a plus or minus sign, respectively. (**G**) CmAreA activated *Cmhyd1* promoter activity cooperating with CmOTam in tobacco leaves. Three plasmids, P*_Cmhyd1_*: LUC, P*_35S_*: CmAreA and P*_35S_*: CmOTam, or two plasmids, P*_Cmhyd1_*: LUC and P*_35S_*: CmAreA, P*_Cmhyd1_*: LUC and P*_35S_*: CmOTam, were cotransformed into tobacco leaves and LUC images were taken 2 d after infiltration. Each column is the mean of more than 4 leaves. Error bars indicate the standard deviation (SD) of replicates. Different letters above the bars indicate significant differences (analysis of variance followed by Duncan’s multiple range test, *p* < 0.05).

**Figure 8 jof-07-00674-f008:**
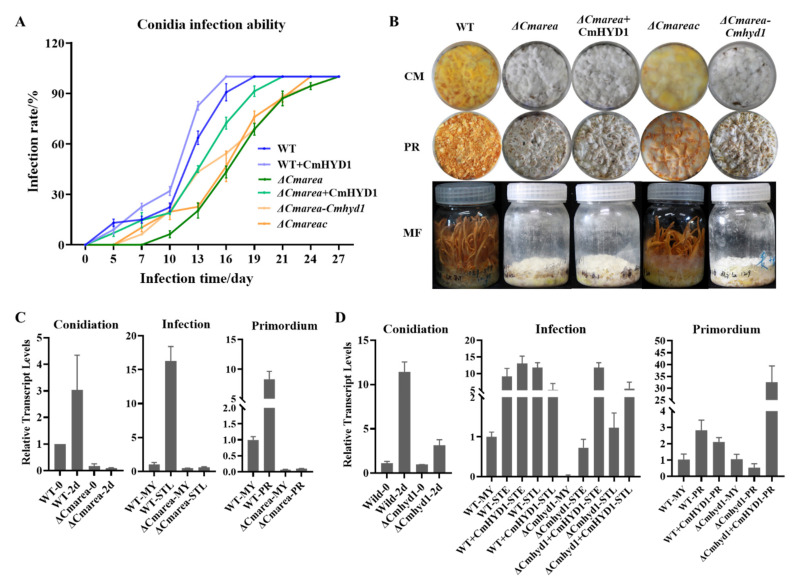
Some phenotypes of the Δ*Cmarea* strain and the relationship between the trancript levels of *Cmhyd1* and *Cmarea*. (**A**) Infection ability of WT and Δ*Cmarea* strains. (**B**) Primordium differentiation and fruiting body development of WT and *Cmarea* mutants. CH: colored hyphae after light irradiation for 4 d; PR: primoduim (stroma < 1 cm); MF: mature fruiting body. “Δ*Cmarea* + CmHYD1” indicates that exogenous CmHYD1 was added to the Δ*Cmarea* strains cultured on wheat media, “Δ*Cmarea**-C**mhyd1*” indicates that the *Cmhyd1* gene was expressed in the Δ*Cmarea* strain. (**C**) The relative transcript levels of *Cmhyd1* in the WT and Δ*Cmarea* strains. Gene transcript levels were based on the darkness treatment (WT-0) and mycelia (WT-MY) of WT strain. (**D**) The relative transcript levels of *Cmarea* in the WT and Δ*Cmhyd1* strains. Gene transcript levels were based on the standard levels of the mycelia in darkness treatment (WT-0), mycelia (WT-MY) and mycelia (WT-MY) of WT strain.

**Figure 9 jof-07-00674-f009:**
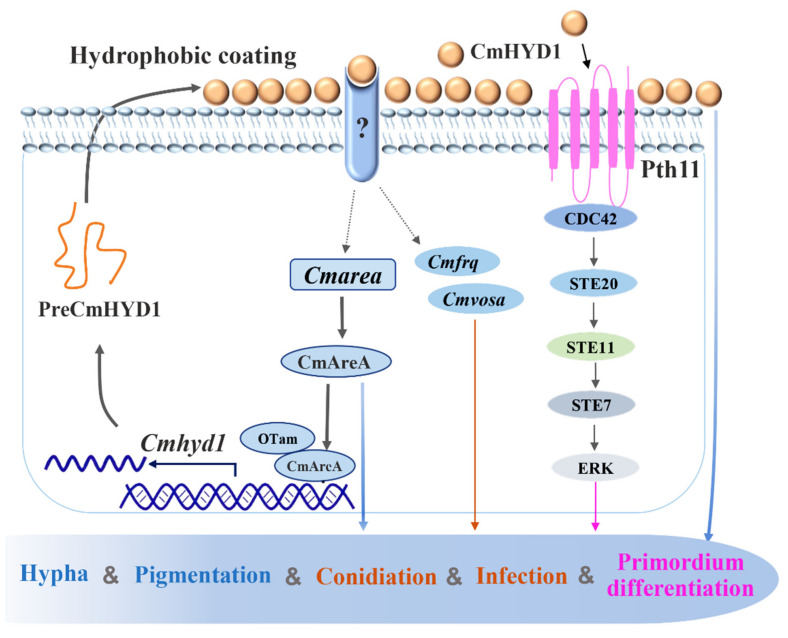
Putative model of *Cmhyd1* regulating hypha growth, pigmentation, conidiation, hydrophobicity and infection as well as primordium differentiation in *Cordyceps militaris*. CmAreA directly activated the transcription of *Cmhyd1*, cooperating with CmOTam. *Cmhyd1* was expressed and CmHYD1 was secreted. CmHYD1 formed a hydrophobic coating for hyphae and conidia to ensure their hydrophobicity. CmHYD1 was involved in conidiation and infection as well as primordium differentiation by regulating the transcription of *Cmfrq*, *Cmvosa* and genes of the MAPK signaling pathway. CmHYD1 also regulated the transcript level of *Cmarea* via a positive feedback mechanism. Dotted arrows indicate the relationship was only confirmed at transcript levels. The “?” indicates unknown protein.

## Data Availability

Not applicable.

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
