# Peer review of "Hydrophobin CmHYD1 Is Involved in Conidiation, Infection and Primordium Formation, and Regulated by GATA Transcription Factor CmAreA in Edible Fungus, Cordyceps militaris"

_jof, 2021, doi:10.3390/jof7080674_

Round 1
Reviewer 1 Report
The study by Li et al. functionally characterized a hydrophobin gene (Cmhyd1) in Cordyceps militaris. This gene was found to be involved in conidiation, pigmentation, infection, primordium formation and stress responses. Its regulation by the transcription factor CmAreA was also addressed. The study is very interesting and scientifically sound. Below, I detail some minor corrections to clarify some points, mostly for non-expert readers.
Line 68: Please explain here why C. militaris is different from species of genera of Beauveria and Metarhizium. I am non experts and maybe I am wrong, but I guess there were some contradictory information about this in past years, so one sentence explaining the differences would be very informative in this regard.
Section 2.7: These fungi infect insect through the cuticle, and thus hydrophobins help with conidia adhesion? If so, why cuticle-bypassing infection?
Section 2.3 and 2.7: Do not start a paragraph with a number. Please write it in full.
Fig. 1B: Could authors explain for non-expert readers what is observed in Fig. 1b and c, in relation to hydrophobicity?
Line 179: This in interesting; but at what concentration was the recombinant hydrophobin added to the wheat medium or mixed with blastospores?
Lines 240-243: What was the fold induction in overexpressing strains? It was assayed by qRT-PCR?
Line 572: than hydrophobins from B. bassiana
Reviewer 2 Report
This manuscript deals with a class-II hydrophobin Cm-hyd1 and a transcription factor Cm-area connected with Cm-hyd1 in Cordyceps militaris, an edible fungus. It is an interesting study but needs some details/explanation before considering further.
- Section 2.3: There is alternative to check conidial hydrophobicity, which should be tested instead of looking at aggregation under microscopy
- Figure 1F, describe the method how conidia production was determined
- Line 245, change as ‘test tube’ in place of tested tube
- The authors should include a figure showing that deltla-Cmhyd1 mycelia are thinner than WT/complemented strains (Figure 1)
- Figure 2 is not at the same magnification, which makes the comparison a bit difficult
- Lines 262-63: What is the pigment present in militaris?
- Why in Figure 1D WT, complemented and overexpressed mutant strains are brownish, and not in Figure 3 CK?
- The data showing the binding of recombinant CmHYD1 to the Cm-hyd1 mutant is lacking.
- What is the role of exogenous CmHYD1 on WT, how it interacts with the WT strain?
- What were the other transcription factors analyzed in addition to Cmarea?
- Being Class-II hydrophobin, comparing militaris hyd1 with A. fumigatus RodA, which is a Class-I hydrophobin, for conidiation is not correct (ref. 38, in the discussion), and the deletion of Class-II hydrophobins in A. fumigatus has no effect on sporulation.
- Class-II hydrophobins mainly function in mycelial breaching, where HYD1 is localized in conidia of militaris if not on the outer surface of their conidia?
Round 2
Reviewer 2 Report
The authors have answered my queries.